# Synergetic Effect of Tumor Treating Fields and Zinc Oxide Nanoparticles on Cell Apoptosis and Genotoxicity of Three Different Human Cancer Cell Lines

**DOI:** 10.3390/molecules27144384

**Published:** 2022-07-08

**Authors:** Mamdouh M. Shawki, Alaa El Sadieque, Seham Elabd, Maisa E. Moustafa

**Affiliations:** 1Medical Biophysics Department, Medical Research Institute, Alexandria University, Alexandria 21561, Egypt; mora337799@gmail.com (A.E.S.); maisamoustafa86@gmail.com (M.E.M.); 2Alexandria University Cancer Research Cluster, Alexandria 21561, Egypt; 3Physiology Department, Medical Research Institute, Alexandria University, Alexandria 21561, Egypt; seham.elabd@alex-mri.edu.eg

**Keywords:** tumor-treating electric fields (TTFs), zinc oxide nanoparticles (ZnO NPs), hepatocellular carcinoma (HepG2), colorectal cancer cell line (HT-29), breast cancer cell line (MCF-7), SRB assay, flow cytometer, comet assay, total antioxidant capacity

## Abstract

Cancer remains a leading cause of death worldwide, despite extraordinary progress. So, new cancer treatment modalities are needed. Tumor-treating fields (TTFs) use low-intensity, intermediate-frequency alternating electric fields with reported cancer anti-mitotic properties. Moreover, nanomedicine is a promising therapy option for cancer. Numerous cancer types have been treated with nanoparticles, but zinc oxide nanoparticles (ZnO NPs) exhibit biocompatibility. Here, we investigate the activity of TTFs, a sub-lethal dose of ZnO NPs, and their combination on hepatocellular carcinoma (HepG2), the colorectal cancer cell line (HT-29), and breast cancer cell lines (MCF-7). The lethal effect of different ZnO NPs concentrations was assessed by sulforhodamine B sodium salt assay (SRB). The cell death percent was determined by flow cytometer, the genotoxicity was evaluated by comet assay, and the total antioxidant capacity was chemically measured. Our results show that TTFs alone cause cell death of 14, 8, and 17% of HepG2, HT-29, and MCF-7, respectively; 10 µg/mL ZnO NPs was the sub-lethal dose according to SRB results. The combination between TTFs and sub-lethal ZnO NPs increased the cell death to 29, 20, and 33% for HepG2, HT-29, and MCF-7, respectively, without reactive oxygen species increase. Increasing NPs potency using TTFs can be a novel technique in many biomedical applications.

## 1. Introduction

Cancer continues to represent a leading cause of mortality worldwide and a significant impediment to expanding life expectancy [1]. The annual number of newly diagnosed cancer cases is projected to rise to 29.5 million. There were 10 million deaths in 2020 with an expectation to increase to 16.4 million per year by 2040, according to a WHO report [2,3]. Cancer, a condition of uncontrolled cell differentiation, has usually been treated by chemotherapy, radiation, and surgery. Despite the remarkable advancements of conventional therapies made in recent decades, in fact, these therapies also introduce a lot of serious side effects, as these treatment methods affect the normal cells as well as the tumor cells [4,5]. Due to the nonselective effect of these therapeutic methods and the development of drug resistance upon long-term use, new treatment alternatives are desperately needed [6].

Although different frequency ranges of alternating electric fields have been demonstrated to have diverse biological effects, the intermediate frequency range of several hundred kHz was assumed to have no therapeutic effect or medical application [7] because currents in this range alternate too quickly to change cell membrane polarization or trigger nerves or muscles (as in the case of direct current or low-frequency alternating fields smaller than 1 kHz) [8]. In comparison to high-frequency fields (MHz range), the intermediate frequency range does not provide enough thermal energy to cause considerable tissue heating [9]. However, since the early 2000s, several studies have demonstrated that low-intensity alternating fields successfully inhibited cancer cell growth, both in vitro (using cell lines derived from melanoma, glioma, lung, prostate, and breast cancer) and in vivo, by interfering with microtubule polymerization during mitosis [10,11,12]. Tumor-treatment fields (TTFs), a physical modality for treating cancer, were developed as a result of these discoveries. TTFs are alternating electric fields in the frequency range of 100–500 kHz, and intensities typically in the range of 1–3 V/cm can exert an antimitotic effect on cells [13]. The Food and Drug Administration (FDA) of the United States approved TTFs for recurrent and newly diagnosed glioblastoma (primary malignant brain tumor) in 2011 and 2015, respectively [14].

Nanobiotechnology is the most rapidly growing discipline of material science that connects several branches of basic sciences. Of the wide range of applications of nanobiotechnology, it is frequently used to reduce cancers with the use of inexpensive inorganic nano-scale materials [15]. When compared to bigger particles of the same composition, nanoparticles (NPs) with a diameter of around 30 nm exhibit radically different mechanical, optical, electrical, magnetic, catalytic, and bioactivity properties [16]. NPs have the potential to easily enter the cells and have an impact on a variety of biological systems [17,18]. Numerous investigations on metal NPs and metal oxide NPs have revealed an increasing influence on cancer treatment [19,20]. Zinc oxide nanoparticles (ZnO NPs) are one of the most important metal oxide NPs in cancer treatment because of their great biocompatibility and biodegradability [21]. ZnO NPs have potent inhibitory effects against tumor cells due to their inherent toxicity, which they achieve by generating intracellular reactive oxygen species (ROS) production and triggering the apoptotic signaling pathway [22], making them a good candidate for anticancer drugs. Previous studies have reported that ZnO NPs have great anticancer properties [23,24,25].

As TTFs are examined in synergy with other traditional antitumor methods as with a chemotherapeutic drug [26] or with ionizing radiation [27]. Recently, a combination of TTFs with other methods as with hyperthermia started to be examined [28]. Moreover, the fear of unexpected toxicity of high concentrations usage of ZnO NPs comes into consideration [29]. Hence, the aim of the present work is to examine the potency of combining TTFs with a sub-lethal dose of ZnO NPs on three human cancer cell lines; hepatocellular carcinoma (HepG2), colorectal cancer cell line (HT-29), and breast cancer cell line (MCF-7).

## 2. Results

### 2.1. The Temperature Measurement Due to TTFs Treatment

A non-effective temperature increase was measured immediately after the TTFs exposure: 1.4 ± 0.1 °C, 0.9 ± 0.2 °C, and only 0.4 ± 0.2 °C for HepG2, HT-29, and MCF-7, respectively. It is concluded that the lethal mechanism of action of TTFs is not a thermal-dependent technique.

### 2.2. The Effect of ZnO NPs Concentration on the Viability of the Tumor Cell Lines Using SRB Assay

In this study, different exponential concentrations of ZnO NPs of 0.1, 1, 10, 100, and 1000 µM have been examined on the various cancer cell lines (HepG2, HT-29, and MCF-7) using SRB as a cytotoxicity assay, SRB micrographs are shown in Figure 1. The SRB results shown in Figure 2 indicate the dose–response curve for each cell line is different for ZnO NPs. We find that 100 µg/mL of ZnO NPs causes death to 83.3% ± 2.9 of HT-29 cells compared with 69.5% ± 0.32 for HepG2 and only 53.3% ± 2.4 for MCF-7. The concentration of ZnO NPs that causes 50% death of the cells (IC50) for HepG2, HT-29, and MCF-7 is 33.9 µg/mL, 38.6 µg/mL, and 12.7 µg/mL, respectively. The target of our work is to use a sub-lethal dose of ZnO NPs, so based on SRB results, we selected 10 µg/mL to be the used dose that causes a cell death percentage of 23.5 ± 1.47, 25.1 ± 1.8, and 22.3 ± 0.3 for HepG2, HT-29, and MCF-7, respectively.

### 2.3. The Effect of Different Treatment Conditions on the Viability of the Tumor Cell Lines Using Flow Cytometer

The viability for control cell lines (without any treatment), treated cells with 10 µg/mL ZnO NPs, treated with (TTFs), and treated with combined 10 µg/mL ZnO NPs and TTFs have been determined by flow cytometry. The flow cytometer images are shown in Figure 3. The flow cytometer results shown in Figure 4 present the percent of gated cells, which indicates a small increased effect of 10 µg/mL ZnO NPs than TTFs on the different cell lines. While highly greater efficiency is obtained upon using the synergetic effect of the two treatments. For HepG2, the normal cells are 85%, 86%, and only 71% for the ZnO NPs group, TTFs group, and the synergistic group, respectively. For HT-29, the normal cells are 85%, 92%, and only 80% for the ZnO NPs group, TTFs group, and the synergistic group, respectively. For MCF-7, the normal cells are 79%, 83%, and only 67% for the ZnO NPs group, TTFs group, and the synergistic group, respectively.

### 2.4. The Effect of Different Treatment Conditions on the Genotoxicity of the Tumor Cell Lines Using Comet Assay

The comet assay is frequently used in genotoxicity assessment, both in vitro and in vivo. In nearly any eukaryotic cell, including cells isolated from tissues, it assesses DNA strand breakage and alkali-labile sites. As shown in Figure 5, the amount of DNA damage, which is the relative amount of fluorescence in the tail compared to the total fluorescence can be visually scored. The results analysis shown in Figure 6 indicates that there is a significant increase in the percent of the tail DNA (genotoxicity) for the TTFs group than in the control. A significant increase in the ZnO NPs group more than that of both the control and TTFs group is observed. In addition, a significant increase in the synergetic group compared to the ZnO NPs group is found for the three cell lines.

### 2.5. The Effect of Different Treatment Conditions on the Total Antioxidant Capacity (TAC) of the Tumor Cell Lines

The potential role of intracellular ROS generation in the cytotoxicity of ZnO was demonstrated in our experiment by measuring TAC since redox reactions at the NP surface are a primary driver. As shown in Figure 7, the TAC of the ZnO NPs group is significantly higher than the control group, on the other hand, the TTFs group is highly significantly lower than the control. The synergetic group has no significant difference compared to the ZnO NPs group. The increase in the produced ROS is a known mechanism in NPs mode of action on tumor cells, and it is found before that the TTFs mechanism of action is not associated with the production of ROS but also decreases its production.

## 3. Discussion

TTFs technique is one of the low-intensity AC fields and the application of an external electric field to human organs is expected to make an obvious change in the cellular processes. As within all the living cells, there are ions, polar molecules, and organelles that may react to the net forces generated by the applied electrical fields during biological processes including DNA replication and cell division [30]. The biophysical and biological effects of TTFs can include forced dipole alignment and dielectrophoresis [31].

TTFs have the ability to disrupt cancer cells mitosis and induce cell cycle arrest ending with mitotic cell death [32]. TTFs can have an interaction with the membrane potential in the dividing tumor cells, causing possible effects on their ionic channels [33]. TTFs can induce also membrane blebbing during telophase, which in turn leads to the formation of abnormal daughter cells and induction of cell death in the following interphase [34]. 

The therapeutic effects of TTFs can include many intracellular mechanisms due to the presence of different charged and polarizable molecules within cells that the TTFs could exert biophysical forces. In addition to the antimitotic effects of TTFs, a multitude of biological processes, including DNA repair, autophagy, cell migration, permeability, and immunological responses, are perturbed by TTFs to elicit anticancer effects [35].

The lethal effect results of TTFs therapy are influenced by treatment duration, electrical field intensity, and electrical field frequency, which varies between cancer types [11,30,36]. This explains our results that the cell death percent obtained in the flow cytometer results are due to fixed TTFs conditions for the three examined cell lines. We found the applied TTFs conditions cause about 14, 8, and 17% of HepG2, HT-29, and MCF-7, respectively. Electrical conditions optimization is suggested for each cancer type and will be a target in our future work.

It was revealed before that gene expression analysis on a variety of non-small cell lung cancer (NSCLC) cell lines treated with TTFs produce not only changes in cell cycle and mitosis-related pathways, but also in DNA damage response pathways [37,38] and this agrees with our work that the % of DNA damage for TTFs group increase by 166%,190%, and 113% for HepG2, HT-29, and MCF-7, respectively, compared to the control.

Not only TTFs lethal mechanism does not include ROS production but also acts to decrease ROS generation by unknown mechanisms till now. These results were achieved in a report for our group before [39]. TAC decreased in the TTFs group by 62%, 75%, and 66% for HepG2, HT-29, and MCF-7, respectively, compared to the control.

Previous researchers agree with our results that the ZnO NPs cytotoxic effect is concentration/dose-dependent as mentioned by R. Wahab et al. [40] for HepG2 and MCF-7, and by Schneider T et al. for HT-29 [41]. The ZnO NPs cytotoxicity depends mainly on NP’s origin, size, exposure time, and the cancer cell type. The ZnO NPs IC50 value for HepG2 cells was reported as 10–15 µg/mL if the ZnO NPs are prepared by a chemical method [42], while IC50 becomes 150 µg/mL if prepared biologically [43]. Similarly, the IC50 value for MCF-7 cells was reported as 40 µg/mL for spherical 40 nm [44] while 6.84 µg/mL for hexagonal 32 nm [45]; although, both are biologically prepared with the same exposure time at 24 h. So, the comparison of the results should take all the work conditions into consideration. Kadhem H. et al. [46] determined IC50 for MCF-7 to be 15.88 μg/mL, which is a close value to our results (12.7 µg/mL) for similar work conditions.

It was reported before that cancer cells can uptake the ZnO NPs by an endocytic route, and because this entry route may vary according to the cell type, its intracellular accumulation differs from one cancer type to another [47]. ZnO dissolves and zinc ions (Zn^2+^) ions are released, the intracellular levels of dissolved Zn^2+^ are highly increased, which can cause severe damage to the electron transport chain. This leads to a massive ROS production intracellularly, which, in turn, causes damage to both mitochondrial and DNA followed by a loss in the balance of protein activities, and finally causes cancer cytotoxicity via an apoptotic signaling pathway [47,48,49]. Sharma et al. published several studies on the genotoxicity of ZnO NPs in a variety of cell systems, they observed DNA damage using the comet assay in the HepG2 human liver cell line and the A-431 human epidermal cell line for cells exposed to 20 µg/mL ZnO NPs for 6 h [50,51]. Compared to our results, the percent of DNA damage for the ZnO NPs group increased by 290%, 255%, and 157% for HepG2, HT-29, and MCF-7, respectively, compared to the control. Other groups also published similar results demonstrating the positive correlation between oxidative stress and DNA damage [52,53]; this also agrees with our current results that TAC increased in the ZnO NPs group by 123%, 107%, and 157% for HepG2, HT-29, and MCF-7, respectively, compared to the control.

In the current work, the high production of ROS in the ZnO NPs group explains the increase in the cell death rate compared to the TTFs group. While the negative correlation between TTFs and ROS production stops the expected increase in TAC in the synergetic group. The synergetic effect of ZnO NPs with TTFs highly significantly increases the death rate for all the tested tumor cell lines compared to each individual treatment. The combination between TTFs and sub-lethal ZnO NPs increased the cell death to 29, 20, and 33% for HepG2, HT-29, and MCF-7, respectively. This potency in the synergetic group was not accompanied by a significant increase in the produced ROS compared to the ZnO NPs group. The synergetic effect magnifies the impact of each individual treatment and eliminates the possible risks of massive ROS production upon using higher ZnO NPs concentrations.

## 4. Materials and Methods

### 4.1. Zinc Oxide Nanoparticles (ZnO NPs)

ZnO nanoparticles with a mean diameter of 30 nm were purchased from Nano Gate, Cairo, Egypt. ZnO NPs characterization, including the absorbance scan (UV–Vis measurement), Fourier Transform Infrared Spectrophotometer (FTIR), X-ray diffraction (XRD), and Scanning electron microscope (SEM), was performed, and the data were consistent with an earlier publication by our group (data not shown) [54]. Zeta potential distribution for ZnO NPs is available in the supplementary material (Appendix A).

### 4.2. Experimental Cell Lines

Three different human cancer cell lines; Hepatocellular carcinoma (HepG2), and Breast cancer cell line (MCF-7) were purchased from the Medical Research Institute, Alexandria University, Egypt, and a colorectal cancer cell line (HT-29) was purchased from the City of Scientific Research and Technological application (SRTA-City), Alexandria, Egypt. Cells were cultured on collagen-coated tissue culture dishes (Sarstedt AG & Co., Nümbrecht, Germany) in a complete medium of high glucose Dulbecco’s Modified Eagle’s Medium (DMEM) with L-Glutamine (Lonza, Brussels, Belgium), supplemented with 10% fetal bovine serum (FBS) (Sigma, Berlin, Germany), 1% penicillin and streptomycin (Biochrom GmbH). Cells were kept under standard cell culture conditions at 37 °C in a humidified CO_2_ incubator containing 5% CO_2_. Cells were grown in a monolayer, and the cell adherence rate during subculture was required to reach 80% confluence. In all experiments, cultures of 80% confluence were used with a maximum passage number of 20–25.

### 4.3. Determination of the Lethal Effect Due to Different ZnO NPs Concentrations

The SRB assay was used to determine the cell viability of each cell line type. In 96-well plates, aliquots of 100 μL for each cell suspension (5 × 10^3^ cells) were incubated in a complete medium for one day. Aliquots of 100 μL media containing ZnO NPs were used at different exponential concentrations (0, 0.1. 1, 10, 100, and 1000 µg/mL) to treat the cells. After 72 h of the treatment, cells were fixed by changing the medium with 150 μL of 10% trichloroacetic acid (TCA) and incubating at 4 °C for 1 h. After removing the TCA solution, the cells were washed five times with distilled water. Aliquots of 70 μL of sulforhodamine B sodium salt (SRB) in 1% (*v*/*v*) acetic acid (SRB solution) were added and incubated at room temperature for 10 min in the dark. Plates were washed 3 times with 1% acetic acid and air-dried overnight. Then, 150 μL of TRIS (10mM) was added to dissolve the protein-bound SRB stain, and the absorbance was measured at 540 nm with a BMGLABTECH^®^-FLUOstar Omega microplate reader (Ortenberg, Germany) [55,56]. The viability % can be obtained using the following equation:(1)$$\mathrm{viability}\;\%=\frac{Sample\;absorbance}{Control\;absorbance}\times 100$$

The experiment on each type of three cancer cell lines was performed 3 times then the average of them was taken.

### 4.4. The Tumor-Treating Field (TTFs) Exposure

The exposure set was extracted from our previous work [39]. Two Ag/AgCl electrodes with a thickness of 1 mm, a width of 1.5 cm, and a height of 3.5 cm; the distance between the two electrodes was 5 cm. Two holes in the lid of the cell culture dish were made; through them, the two electrodes were passed and then fixed. Each electrode was connected to one output terminal of the function generator (CALTEK, CA1640P-02 function generator/counter, serial number: 06mg0676, Boston, MA, USA). The exposure was performed by placing the electrodes through the cell culture media. Application of 2 V/cm AC for 5 min was performed for the different cell lines. The temperature of each cell line was measured before and immediately after TTFs using a digital thermometer (Infrared Thermometer CK-T1501, Shenzhen, China).

### 4.5. Analysis of Cell Apoptotic State by Flow Cytometry

The percent of gated cells (cell cycle analysis) was performed by determining the quantity of DNA content. This was preceded by staining the cells with a fluorescent DNA-binding dye such as propidium iodide (PI), which was added to permeabilized single cells. PI will bind to DNA, and during flow cytometry, the DNA will emit a fluorescent signal, which varies depending on the amount of DNA in the cell. After each treatment condition, cells were harvested and washed with PBS. Cells were fixed and permeabilized in 80% ethanol for 1 h. Then, the fixed cells were washed with phosphate buffer solution (PBS) followed by incubation with RNase I (10 μg/mL) at 37 °C for 30 min. Finally, the cells were re-suspended in 50 μg/mL PI at 4 °C for 30 min, and then detected using a FACScan flow cytometer (Becton Dickinson, Franklin Lakes, NJ, USA). Data were acquired and recorded using the BD CellQuest™ Pro software. Fluorescent Count Bright counting beads were added to samples to enable determinations of absolute cell numbers, and changes in PI staining were used to quantify cell death [57]. Nanoparticles were excluded from the analysis based on the absence of fluorescence signal and light forward scatter and side scatter characteristics, and samples were analyzed using a Caliber flow cytometer.

### 4.6. Genotoxicity Determination (Comet Assay)

The comet assay (single-cell gel electrophoresis) is a simple method for measuring deoxyribonucleic acid (DNA) strand breaks in eukaryotic cells. Each treated group was mixed with low melting agarose before embedding on the comet slide. The cells were lysed in prechilled lysis buffer for one hour, followed by denaturation in alkaline running buffer. The comet slide was then placed on a horizontal electrophoresis tank, and run at 25 V for 20 min. Cells were placed inside neutralizing buffer before being subjected to dehydration with 70% ethanol. Dried cells were stained with SYBR Green before viewing under a Zeiss Axioplan 2 imaging fluorescence microscope. The 100 cells for each slide or slide sub-area, were classified according to the 5 types of damage and can be measured with the visual scoring method [58]. Visual scoring, automated, and semi-automated image analysis methods are capable of detecting a significant level of damage at the lowest concentration of each agent. Visual scoring systematically overestimates low levels of damage compared with computerized image analysis; on the other hand, heavily damaged comets are less efficiently detected with image analysis. Overall, the degree of agreement between the scoring methods was within acceptable limits [59].

### 4.7. Determination of Reactive Oxygen Species (ROS)

To estimate the generation of ROS and antioxidant activity, this was performed by using TAC kit (ab65329). TAC was measured via a SET (single electron transfer) mechanism. In SET mechanisms, the antioxidant provides an electron to the free radical and itself then becomes a radical cation [60]. The antioxidant color of the samples was calculated by detecting the absorbance value at 490 nm.

### 4.8. Statistical Analysis

Experiments were repeated in triplicate, and the results are presented as mean ± standard deviation (S.D). ANOVA was used to evaluate the difference between multiple groups. Significant differences between experimental groups were determined using a two-tailed Student’s *t*-test (Excel 2013 Microsoft Redmond, WA, USA). Significantly different results were denoted at *p* ≤ 0.05.

## 5. Conclusions

A single exposure to TTFs can work as a new cancer therapeutic technique with a death percentage of 8–17% according to the cancer type. TTFs mechanism of action does not include ROS production. ZnO NPs have a great anti-cancer effect, its IC50 varies from one cancer type to another. Its lethal effect is mainly due to ROS generation. The synergetic effect of TTFs and sub-lethal ZnO NPs increases the lethal percentage to be about 20–33% according to the cancer type without an increase in ROS production. The results of this research can lead to the use of smaller concentrations of NPs as well as improving the TTFs technique against different cancer cell types. Increasing the potency of the NPs using TTFs can be a novel technique in many biomedical applications. Decreasing the number of used NPs has environmental and industrial importance.

## Figures and Tables

**Figure 1 molecules-27-04384-f001:**
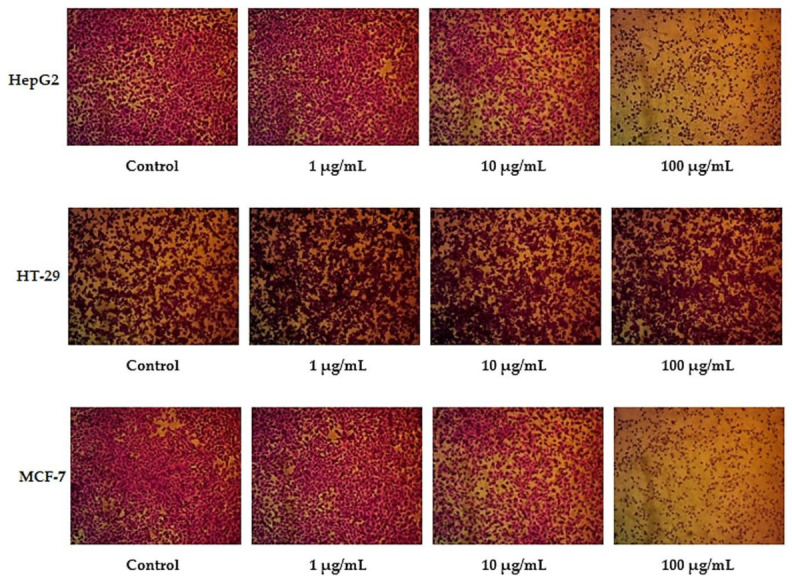
SRB assay micrographs for HepG2, HT-29, and MCF-7 cell lines at different ZnO NPs concentrations.

**Figure 2 molecules-27-04384-f002:**
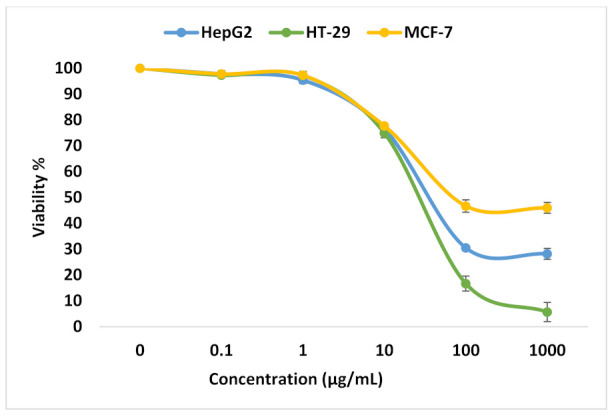
Analysis for SRB assay. Viability % for HepG2, HT-29, and MCF-7 cell lines at different ZnO NPs concentrations.

**Figure 3 molecules-27-04384-f003:**
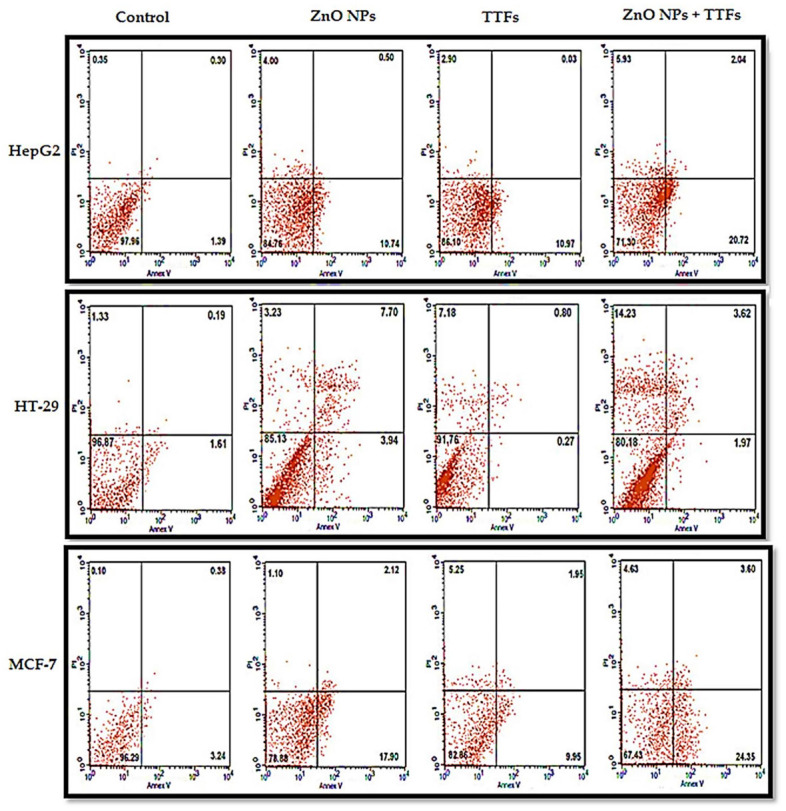
Flow cytometer micrographs of annexin V and PI staining tumor cell lines (HepG2, HT-29, and MCF-7) after treatment with TTF, ZnO NPs, or both, and untreated control was referred as the control. Percentages shown in the upper left, upper right, lower, left, and lower right quadrants are percentages of gated cells showing necrosis, late apoptosis, viability, and early apoptosis, respectively.

**Figure 4 molecules-27-04384-f004:**
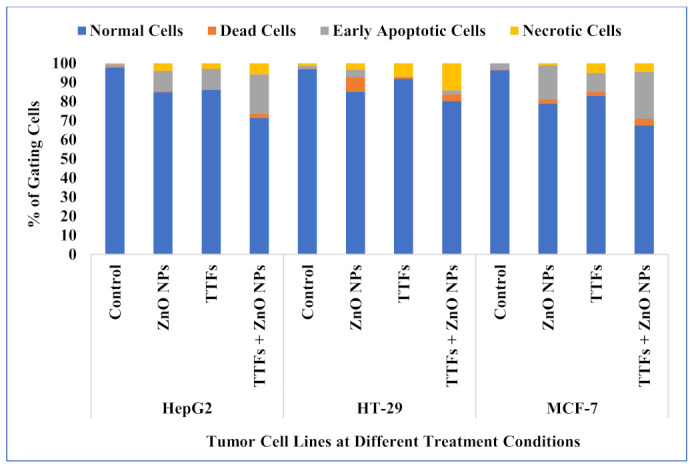
Flow cytometer analysis for HepG2, HT-29, and MCF-7 cell lines at the different treatment conditions.

**Figure 5 molecules-27-04384-f005:**
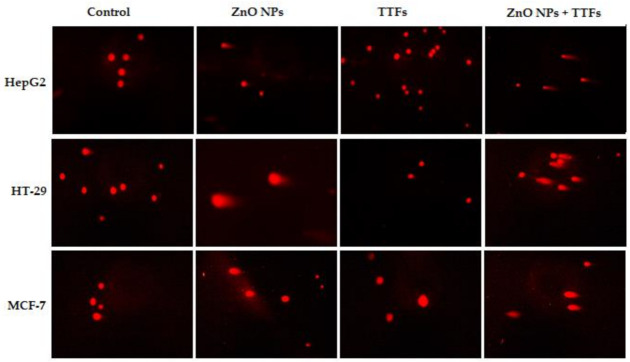
Fluorescence micrograph representing nuclei of HepG2, HT-29, and MCF-7 cell lines after the comet assay. Control nuclei of untreated cells appear intact with no detected DNA damage. Nuclei of tumor cells treated with a combination of TTFs and ZnO NPs appear to have more damage than with TTFs or ZnO NPs alone.

**Figure 6 molecules-27-04384-f006:**
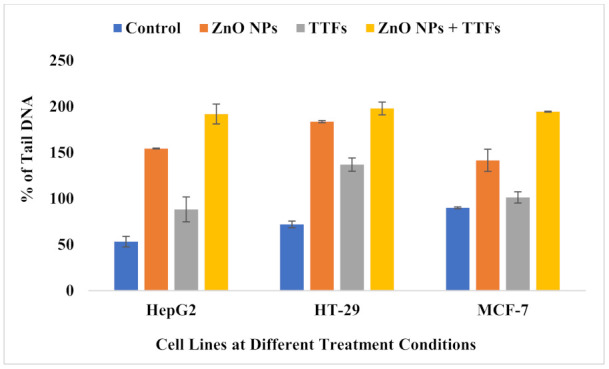
Bar graph showing the percentage of tail DNA damage in the nuclei of tumor cells of untreated control, ZnO NPs treated, TTFs treated, and combination-treated cells. Data are shown as mean ± SD for three independent experiments.

**Figure 7 molecules-27-04384-f007:**
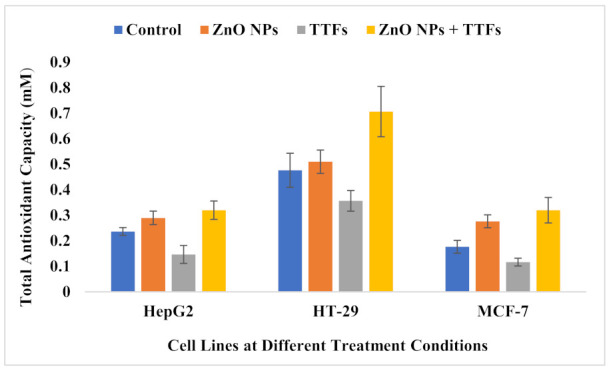
Total antioxidant capacity (mM) for HepG2, HT-29, and MCF-7 cell lines at the different treatment conditions. Data are shown as mean ± SD for three independent experiments.

## Data Availability

The data presented in this study are available upon request from the corresponding author.

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
