# Peer review of "Synergetic Effect of Tumor Treating Fields and Zinc Oxide Nanoparticles on Cell Apoptosis and Genotoxicity of Three Different Human Cancer Cell Lines"

_molecules, 2022, doi:10.3390/molecules27144384_

Round 1

Reviewer 1 Report

This study about a combined treatment of ZnO NPs with TTFs is very promising and interesting. Although this study is of a high quality, there are some points that have to be addressed.

1) In the method section it is mentioned that the ZnO nanoparticles are purchased from Nano Gate, Egypt. For the characterization ref. 54 was cited. In this article that ZnO NP were synthesized by the authors and not purchased. I doubt that the UV-VIs, FT-IR, XRD and TEM are really the same for the purchased nanoparticles.

2) Did the authors perform any DLS measurements? How about the stability of the nanoparticles in biological media?

3) The authors mentioned that the TTFs disrupt the cell cycle mitosis and causes cell cycle arrest. Did the authors measure the cell cycle phases with the treatment? Do the ZnO NP cause any cell cycle arrest by themselves without any TTFs? I recommend to do a cell cycle analysis for the cells with/without NPs and TTFs respectively.

4) In line 250 the authors mentioned the intracellular accumulation of the ZnO NPs differ from cancer type to another. Did the authors measure the amount of nanoparticles that were taken up by the different cell line? Are there any differences in NP uptake? The authors maybe have to consider this in the discussion.

Author Response

Respectable Ms. Charlotte Yu

Respectable Reviewer

With respect

Mamdouh M Shawki

Reviewer 2 Report

the paper titled 'Synergetic Effect of Tumor Treating Fields and Zinc Oxide Nanoparticles on Cell Apoptosis and Genotoxicity of Three Different Human Cancer Cell Lines' described the synergistic effect of TTF and ZnO NOPs on three types of cancer cells. some issues should be dissolved prior to publication.

1) the authors said the ZnO NPs exhibit biocompatibility in the abstract, in this case, why this kind of NPs was chosen for combination therapy? 

2) the 10ug/mL was chosen for sub-lethal dose based on the SRB results, but the difference in concentration (1, 10, 100) is too huge, so appropriate concentraion should be considered carefully.

3) the quality of some figures should be improved, i.e. figure 2, 4, 6, 7, the lines in the figures can be deleted.

Author Response

(The authors gave the same response as above.)

Reviewer 3 Report

Shawki et al. investigated an interesting research titled Synergetic Effect of Tumor Treating Fields and Zinc Oxide Nanoparticles on Cell Apoptosis and Genotoxicity of Three Different Human Cancer Cell Lines. Although the research is scientifically sound and innovative, it must be revised to solve some important issues before it can be accepted for publication in the molecules.

 1.      Please double-check and maintain the following terms in the manuscript for consistency [For example, space between numbers and unit, full form and abbreviated form (min/minutes)]

2.      Maintain the abbreviation throughout the thesis; it is not essential to abbreviate again after it has been abbreviated [For example, total antioxidant capacity (TAC)].

3.      Abstract – When compared to methods, the results section falls short. Make it better by including numerical results of the important findings. Additionally, add future perspectives to the conclusion.

4.      Introduction – Line 73, Many publications – Change this sentence to “Previous studies have reported………….. Authors may need to include a few more recent references to support that statement in line 73–74 (https://doi.org/10.2147%2FIJN.S328135). Line 79, start with “Hence, the aim of the present study was to examine…….”

5.      The results section has satisfied myself. However, the discussion section might be improved by including the most recent references. Because the discussion section also has some results (numerical values) that are similar.

6.      Line 237-240, the sentence is not clear and not appropriate. Please rewrite it.

7.      The methodology section also well written. However, it's important to keep some procedures as short as possible (For example, 4.3, 4.5 and 4.6).

8.      In conclusion – The author should strengthen the work's novelty. The findings/insights should be used to support this section. The importance of the study should be emphasized by the author. Future viewpoints also need to be expanded.

Author Response

(The authors gave the same response as above.)

Round 2

Reviewer 1 Report

The author answered all my questions. I would recommend to include the DLS measurements and zeta potential measurements to the supplementary material.

Reviewer 2 Report

all issues have be dissolved, and it is acceptable by Molecules.

Reviewer 3 Report

The authors took the reviewer comments into account and revised their manuscript as needed. I therefore recommend that this be published in its current form in Molecules Journal.